# Position and Attitude Determination in Urban Canyon with Tightly Coupled Sensor Fusion and a Prediction-Based GNSS Cycle Slip Detection Using Low-Cost Instruments

**DOI:** 10.3390/s23042141

**Published:** 2023-02-14

**Authors:** Bálint Vanek, Márton Farkas, Szabolcs Rózsa

**Affiliations:** 1Institute for Computer Science and Control, Eötvös Lóránd Research Network, Kende u. 13-17, H-1111 Budapest, Hungary; 2Department of Geodesy and Surveying, Faculty of Civil Engineering, Budapest University of Technology and Economics, Műegyetem rkp. 3, H-1111 Budapest, Hungary

**Keywords:** GNSS-INS fusion, cycle slip detection, low-cost sensor integration, urban canyon environment, moving baseline GNSS

## Abstract

We present a position and attitude estimation algorithm of moving platforms based on the tightly coupled sensor fusion of low-cost multi baseline GNSS, inertial, magnetic and barometric observations obtained by low-cost sensors and affordable dual-frequency GNSS receivers. The sensor fusion algorithm is realized by an Extended Kalman Filter and estimates the states including GNSS receiver inter-channel biases, integer ambiguities and non-GNSS receiver biases. Tightly coupled sensor fusion increases the reliability of the position and attitude solution in challenging environments such as urban canyons by utilizing the inertial observations in case of GNSS outage. Moreover, GNSS observations can be efficiently used to mitigate IMU sensor drifts. Standard GNSS cycle slips detection methods, such as the application of triple differences or linear combinations such as Melbourne–Wübbena combination and the phase ionospheric residual extended TurboEdit method. However, these techniques are not well suited for the localization in quickly changing environments such as urban canyons. We present a new method of tightly coupled sensor fusion supported by a prediction based cycle slip detection technique, applied to a GNSS setup using three antennas leading to multiple moving baselines on the platform. Thus, not only the GNSS signal properties but also the dynamics of the moving platform are considered in the cycle slip detection. The developed algorithm is tested in an open-sky validation measurement and two sets of measurement in an urban canyon area. The sensor fusion algorithm processes the data sets using the proposed prediction-based cycle slip method, the loss-of-lock indicator-based, and for comparison, the Melbourne–Wübbena and the TurboEdit cycle slip detection methods are also included. The obtained position and attitude estimation results are compared to the internal solution of raw data source GNSS receivers and to the observations of a high-accuracy GNSS/INS unit including a fiber optic gyro. The validation test confirms the proper cycle slip detection in an ideal environment. The more challenging urban canyon test results show the reliability and the accuracy of the proposed method. In the case of the second urban canyon test, the proposed method improved the integer ambiguity resolution success rate by 19% and these results show the lowest horizontal and vertical coordinate distortion in comparison of the linear combination and the loss-of-lock-based cycle slip methods.

## 1. Introduction

High-accuracy localization is one of the key components in the penetration of autonomous vehicles in road transport. Decision mechanisms rely heavily on the outputs of different localization sensors such as Global Navigation Satellite Systems (GNSS) receivers, inertial navigation systems (INS), magnetometer, barometer, odometer sensors, LIDAR sensors, or stereo cameras used for envirionmental perception as well as onboard databases such as High-Definition (HD) Maps. Localization aims at providing safe, redundant, and high-accuracy positions and attitudes to support the autonomous decision making. This can be achieved through the high-level integration of the aforementioned observations. These sensors have different advantages and disadvantages. LIDARs and cameras can provide high-accuracy relative positioning but they need large amount of supporting data and cause a significant computational load for absolute positioning such as simultaneous localization and mapping (SLAM). GNSS receivers provide accurate absolute positioning data, but they are sensitive to the measurement environment, for example, RF interference, multipath, signal losses caused by urban canyons or tunnels. Furthermore, even modern GNSS receivers provide position samples at a relatively low rate (typically up to 10 Hz) due to the time required for the measurement and processing steps. The inertial and magnetic sensors can support localization with high-rate observations, where the GNSS measurements are subject to outages or higher uncertainty. On the other hand, inertial and magnetic sensors suffer from errors such as bias, drift or scale factor, which prohibit the use of these sensors alone for long-term navigational tasks. It is well known that integration of different sensors could provide a solution for the redundant and safe navigational tasks [1]. The key to high-accuracy positioning is the ability of processing carrier-phase measurements and the resolution of the integer ambiguities. The carrier-phase measurements may suffer from cycle slip errors, when the carrier-phase lock is lost and re-initialized. The undetected cycle slips can cause particularly large positioning errors, especially in weak satellite reception environments, for example, during satellite–receiver line of sight losses or large amounts of signal losses. The literature on cycle slip detection and repair spans almost three decades. Lipp and Gu developed an algorithm which uses a dynamical extension of a Doppler prediction method, which re-transforms the receiver speed into approximate Doppler values [2]. This method was only tested in a simulated environment.

The multi-frequency technology made it possible to create different linear combinations of the measurements coming from the same satellite. These methods do not need any aid from different sensors. Melbourne [3] and Wübbena [4] used simultaneously the wide-lane combinations of the carrier-phase and the narrow-lane combinations of the pseudorange measurements to estimate the integer ambiguities. Blewitt [5] developed the TurboEdit (TE) method which uses the Melbourne–Wübbena (MW) combination and the ionospheric combinations of the measurements to detect and repair the cycle slips. The method was further developed by Xu et al. [6] to detect small cycle slips with low-cost equipment, such as the cellphone environment. The development of GNSS technology made it possible to use triple frequency-based linear combinations for cycle slip detection, such as in the work of De Lacy et al. [7]. They developed a method using undifferenced GNSS observations, what was tested in simulated and real low-multipath environment. These techniques are sensitive to the continuity of the multi-frequency measurements and the noise of the pseudorange observations. The aforementioned conditions limit their applicability in urban canyon environments, where signal loss and multipath effects are extremely common.

Use of differenced carrier-phase measurements is another widespread method for cycle slip detection. Differencing technique is the principle of the kinematic (Real-Time Kinematic (RTK), Post-Processed Kinematic (PPK)) positioning methods. These differencing steps eliminate the common errors of the GNSS positioning between the base (permanent or moving base) and the rover receiver of the baselines. Differencing can be performed between the receivers of the baselines, which reduces satellite related delays (satellite clock bias, relativistic delay, group delay, atmospheric delays). However, differencing is also used on the same baseline between satellite measurements, which reduces the receiver-related errors (receiver clock bias, hardware delays). Time differentiating reduces errors that change slowly over time or has negligible dynamic effects. The literature names these differences according to the order of implementation, in general, the between-receiver, between-satellite, between-time differences are called single (SD), double (DD) and triple differences (TD), respectively. Lee et al. [8] used the double differenced measurements and the additional information provided by an INS, with the cumulative-sum statistical technique for cycle-slip detection.

Yoon et. al developed a cross-ratio detection methodology, which uses sequentially differentiated double differential carrier-phase measurements and moving averaged Doppler data [9].

Time differencing the carrier-phase measurements reduces the effects of the dynamics. Kim et al. presented their algorithm [10], which detects cycle slips by comparing the time differentiated satellite differences of the carrier-phase (TDCP) measurements obtained from the GPS satellites with the range estimated by the integrated navigation solution applying satellite geometry. The method detected one-cycle slips successfully by using single-frequency GPS receiver measurements when there were more than four visible satellites. Kim et al. developed a navigation system using the time-differenced carrier-phase and inertial measurements [11] and extended it with a cycle slip detection method, which uses the estimated relative position difference between the INS and TDCP measurements. The estimation performed well in deep urban canyons [12], with sections where the visible satellite number decreased to 2–6 per constellation.

The linear combination based methods are functioning well in rather ideal receiving conditions where the availability of GNSS signals is stable for the different measurement frequencies. The differencing techniques need additional information from other sensor measurements about the dynamics to compensate, but they can be used in rapidly changing measurement environments.

The proposed prediction based cycle slip (PBCS) detection method uses the triple differenced (differentiating between receivers, satellites and epochs) carrier-phase measurements. The triple (time)-differenced carrier-phase measurements are used to detect cycle slips in static measurement setup. However, this differencing approach does not work in moving receiver scenarios, because the dynamic effects appear in the measurements. To reduce the dynamic effects, we use the information from the localization estimation algorithm. We also calculate the triple-differenced carrier-phase measurement vector using the carrier-phase measurement model and the predicted state vector. The comparison between the predicted and the real measurement infers the existence of a cycle jump. This method is frequency-independent, and can be used on single-frequency measurement as well. The cycle slip detection is defined in the ambiguity space, all of the modeled and estimated variables are projected to there. Another advantage of this representation is that it can be used also for non stationary rovers and it is also applicable to moving baseline cycle slip detection.

The proposed detection method is applied in PPK approach with a tightly coupled filter integration setup, where the raw observations of the GNSS receivers are processed in the estimation algorithm, in contrast to the traditional loosely coupled integration method [1] where processed PVT data are used from the GNSS receiver. The presented Extended Kalman Filter (EKF) based algorithm uses multi-baseline, multi-constellation, multi-frequency carrier-phase, pseudorange, Doppler GNSS measurements and inertial, magnetic, barometric sensor data to estimate the navigational and sensor error states. The prediction-based cycle slip detection method is tested on real observations made in open-sky and urban canyon environments. The result of the proposed PBCS, the more conventional loss-of-lock indicator (LLI) and linear combination based MW and TurboEdit (TE) methods are compared. The sensor setup of the measurement platform also allows to test all the implemented cycle slip detection methods for the moving baseline based attitude estimation, on which the EKF algorithm relies heavily.

The structure of the paper is as follows. The first section presents the implemented sensor models. This is followed by a description of the multi-constellation, multi-baseline-based tightly coupled estimation algorithm. The fourth section presents an overview of the used cycle slip detection methods, and describes the proposed prediction-based method which is followed by the presentation of the measurement platform. The open-sky and the urban canyon tests and the corresponding position and orientation estimation results are detailed in Section 6, Section 7 and Section 8. The paper is concluded with the discussion of the presented method and the achieved results.

The method has been successfully validated by an open-sky test. This will be followed by urban canyon tests to assess the reliability and accuracy of the method. The results show that the PBCS method-based estimation improves the integer ambiguity resolution success rate by 19% in position estimation and 10% in attitude estimation. The application of the method reduces the position errors, the magnitude of position jumps and the attitude angle uncertainties in the more challenging, second urban canyon measurement case, where the performance of the linear combination-based detection methods is low.

## 2. Sensor Models

The estimation algorithm relies on the observations models of the respective sensors. This section describes the applied observation equations for each sensor.

### 2.1. GNSS Receivers

The estimation algorithm uses the pseudorange, the carrier-phase, and the Doppler-delay measurements of the permanent base, the moving base and the moving rover receivers (Figure 1).

The non-differenced measurement Equations (Equation 1)–(Equation 3) are based on [13].
(1)$$\begin{array}{cc} \varrho_{r}^{s} & =\rho_{r}^{s}+c\left(\delta_{r}-\delta^{s}-\delta^{rel}-\delta^{GD}\right)+I+T+\eta_{\varrho } \end{array}$$
(2)$$\begin{array}{cc} \lambda^{s}\phi_{r}^{s} & =\rho_{r}^{s}+c\left(\delta_{r}-\delta^{s}-\delta^{rel}-\delta^{GD}\right)-I+T+\lambda^{s}\left(ICB_{r}+N^{s}\right)+\eta_{\phi } \end{array}$$
(3)$$\begin{array}{cc} \lambda^{s}d_{r}^{s} & =-E_{r}^{s}\left(v_{r}-v^{s}\right)+c\left(\dot{\delta_{r}}-\dot{\delta^{s}}\right)+\eta_{d} \end{array}$$
where $\rho_{r}^{s}$ is the norm of receiver ($x_{r}$)–satellite ($x^{s}$) position vector (Equation 4), $E_{r}^{s}$ is the line of sight vector (Equation 5)
(4)$$\begin{array}{cc} \rho_{r}^{s} & =∥x_{r}-x^{s}∥ \end{array}$$
(5)$$\begin{array}{cc} E_{r}^{s} & =\frac{x_{r}-x^{s}}{\rho_{r}^{s}}, \end{array}$$
*c* is the speed of light, $\delta_{r}$ and ${\dot{\delta }}_{r}$ are the clock bias and drift of the receiver, $\delta^{s}$ and ${\dot{\delta }}^{s}$ are the clock bias and drift of the satellite, $\delta^{rel}$ and $\delta^{GD}$ are the relativistic and the group delay of the satellite. The ionospheric and tropospheric delays are marked with *I* and *T*. The inter-channel bias error ($ICB_{r}$) associated with the GLONASS system is given by the carrier-phase equation, this term is equal to zero in the case of other constellations. The non-differenced integer ambiguity is $N^{s}$ and $\lambda^{s}$ is the wavelength of the given satellite measurement. The speed of the given receiver and the satellite are $v_{r}$ and $v^{s}$. The pseudorange, the carrier-phase, and the Doppler-delay measurements are marked with $\varrho_{r}^{s}$, $\phi_{r}^{s}$, and $d_{r}^{s}$, respectively. The not modeled error terms are included in the respective η terms. The proposed cycleslip method and the estimation algorithm use the differentiated GNSS measurements, which are the following. The single differencing (SD) is realized between the common satellites of the receivers on the given baseline, thus, the satellite correlated errors (satellite clock bias and drift, atmospheric delays) can be reduced
(6)$$\begin{array}{c} \phi_{rb}^{j}=\phi_{r}^{j}-\phi_{b}^{j}. \end{array}$$ A pivot satellite (usually with the highest elevation) by constellations and frequencies is chosen and its value is deducted from the others. This gives the receiver correlated error-free (receiver clock bias and drift, inter-channel bias, hardware delay) double-differenced (DD) values
(7)$$\begin{array}{c} \phi_{rb}^{jk}=\phi_{rb}^{j}-\phi_{rb}^{k}. \end{array}$$ The triple differencing (TD) is realised between the DD values in time
(8)$$\begin{array}{c} \phi_{rb}^{jk}{|}_{t-1}^{t}=\phi_{rb}^{jk}{{|}_{t}-\phi_{rb}^{jk}|}_{t-1}. \end{array}$$ The TD measurements include the position changes and the cycle slips in case of carrier-phase measurements.

### 2.2. Inertial, Magnetic and Barometric Sensors

The position and attitude estimation method uses the raw acceleration and gyroscope data of the INS system, too. The equation of the measured acceleration (a) is the one given by [14] transformed to the quaternion representation
(9)$$a={R_{\mathrm{body}}^{\mathrm{ecef}}}^{T}A_{\mathrm{MB}}-\mathrm{skew}\left(\omega \right)\mathrm{skew}\left(\omega \right)b_{{\mathrm{MB}}_{\mathrm{body}}}^{\mathrm{INS}}+\;\;\;\;\;\;\;\;\;\;\;\;\;\;\;\;\;\;\;\;\;\;\;\;\;\;\;\;\;\;\;\;\;\;+b_{a}-{R_{\mathrm{body}}^{\mathrm{nav}}}^{T}\left[\begin{array}{c} 0\\ 0\\ g \end{array}\right]+{R_{\mathrm{body}}^{\mathrm{ecef}}}^{T}\left(2\mathrm{skew}\left(\left[\begin{array}{c} 0\\ 0\\ \Omega_{E} \end{array}\right]\right)V_{MB}\right)+\eta_{a},$$
where $R_{\mathrm{body}}^{\mathrm{ecef}}$ is the rotation matrix from body to Earth-Centered Earth-Fixed (ECEF) coordinate system, $A_{\mathrm{MB}}$ is the estimated acceleration of the moving base receiver, Ω is the estimated angular velocity, $b_{{\mathrm{MB}}_{\mathrm{body}}}^{\mathrm{INS}}$ is the lever arm between the moving base antenna and the INS sensor in body coordinate system, $b_{a}$, is the estimated accelerometer sensor bias, $R_{\mathrm{body}}^{\mathrm{nav}}$ is the body-to-navigational transformation matrix, *g* is the local gravity, $\Omega_{E}$ is the angular velocity of Earth’s rotation and $V_{MB}$ is the velocity state vector of the moving-base receiver. The equation of the measured angular velocity (ω) is
(10)$$\begin{array}{c} \begin{array}{cc} \omega \; & =\Omega +b_{\omega }+R_{\mathrm{body}}^{\mathrm{ecef}}T\left[\begin{array}{c} 0\\ 0\\ \Omega_{E} \end{array}\right]+{R_{\mathrm{body}}^{\mathrm{nav}}}^{T}\omega_{{\mathrm{ecef}}_{\mathrm{nav}}}^{\mathrm{nav}}+\eta_{\omega }, \end{array} \end{array}$$
where Ω is the estimated angular velocity vector, $b_{\omega }$ is the gyroscope sensor bias, $\omega_{{\mathrm{ecef}}_{\mathrm{nav}}}^{\mathrm{nav}}$ is the angular rates of the navigation coordinate system with respect to the ECEF coordinate system [14]. The magnetometer measurement (m) model equation consists of terms as the navigational-to-body transformation matrix ($R_{\mathrm{body}}^{\mathrm{nav}}{}^{T}$), the local magnetic field vector (M) and the magnetometer bias ($b_{m}$)
(11)$$\begin{array}{c} \begin{array}{c} m=R_{\mathrm{body}}^{\mathrm{nav}}{}^{T}M+b_{m}+\eta_{m}. \end{array} \end{array}$$ The barometric sensor model (*b*) is the following, based on [15]:(12)$$\begin{array}{c} b=P_{b}{\left[\frac{T_{b}+\left(h\left(X_{\mathrm{MB}}\right)-h_{b}\left(X_{\mathrm{MB}}\right)-b_{b}\right)L_{b}}{T_{b}}\right]}^{\frac{-gM_{E}}{R^{*}L_{b}}} \end{array}$$
where $P_{b}$ is the reference pressure, $T_{b}$ is the reference temperature, $L_{b}$ is the temperature lapse rate, $h(X_{\mathrm{MB}})$ is the height of the moving base receiver’s antenna ($X_{\mathrm{MB}}$), $h_{b}\left(X_{\mathrm{MB}}\right)$ height of reference level, $b_{b}$ is the barometer sensor bias term, $R^{*}$ is the universal gas constant, g is the gravitational acceleration and $M_{E}$ is the molar mass of Earth’s air. The η terms in the equations are the non-modelled errors, as already described at GNSS models.

## 3. The Estimation Algorithm

The EKF base position and attitude estimation algorithm is introduced in this section including the state vector, the prediction and the update steps [16,17]. The estimated states (x), which are linked to the navigation data and the different sensor errors are

Position ($X_{\mathrm{MB}}$), velocity ($V_{\mathrm{MB}}$) and acceleration ($A_{\mathrm{MB}}$) of the Moving Base antenna in Earth-Centered Earth-Fixed (ECEF) Coordinate system;Orientation quaternions (q), angular velocities (Ω);Accelerometer bias error ($b_{a}$), gyroscope bias error ($b_{\omega }$), magnetometer bias error ($b_{m}$), barometer bias error ($b_{b}$);Local magnetic field (M);GNSS receiver clock biases for every receiver ($\delta_{i}^{GPS},\delta_{i}^{GAL},\delta_{i}^{GLO},\delta_{i}^{BDS}$);GNSS receiver clock drifts for every receiver (${\dot{\delta }}_{i}^{GPS},{\dot{\delta }}_{i}^{GAL},{\dot{\delta }}_{i}^{GLO},{\dot{\delta }}_{i}^{BDS}$);Single-differenced GLONASS system related receiver inter-channel biases for every baseline (${\mathrm{ICB}}_{ri-j}$);Single-differenced integer ambiguities for every baseline and every satellite ($N_{i-j}$).

The structure of the EKF algorithm is depicted in Figure 2. The first step is the measurement data availability check, which determines the used update steps in the current epoch. This is followed by the EKF prediction of the states (${\hat{x}}_{t}$) and their covariance matrix (${\hat{P}}_{t}$) from the previously updated epoch ($x_{t-1}$, $P_{t-1}$)
(13)$$\begin{array}{cc} {\hat{x}}_{t} & =F_{t-1}^{t}x_{t-1} \end{array}$$
(14)$$\begin{array}{cc} {\hat{P}}_{t} & =F_{t-1}^{t}P_{t-1}{F_{t-1}^{t}}^{T}+Q_{t}. \end{array}$$ The $F_{t-1}^{t}$ is the state transition matrix, which is the partial derivative of the dynamic model based on slowly changing accelerations and angular velocities, and $Q_{t}$ is the process noise covariance matrix. If there are newly available GNSS measurements, the algorithm pre-processes them. The satellite-related data (satellite position and velocity, satellite clock and hardware delays, atmospheric delays) and the approximate receiver data (position, velocity, clock biases) are calculated by the least squares estimation of the non-differentiated pseudorange measurements. The following step is the cycle slip detection, which defines whether the ambiguity states need to be re-initialised. This will be explained in more detail in a later section. The innovation residual vectors and the Jacobian matrices are generated using the measurement models and measured data. The EKF algorithm uses the double-differenced carrier-phase, pseudorange and Doppler-delay measurements of the given receivers of the baselines. The EKF algorithm also calculates innovation residual vectors of the inertial, magnetic and barometric sensors for the update step. All of the sensor models will be explained in the following section. To calculate the updated states, the Kalman gain is defined as: ($K_{t}$)
(15)$$\begin{array}{cc} K_{t} & ={\hat{P}}_{t}H_{t}^{T}{\left(H_{t}{\hat{P}}_{t}H_{t}^{T}+R_{t}\right)}^{-1}, \end{array}$$
where $H_{t}$ is the Jacobian matrix from the observation equations and $R_{t}$ is the measurement noise matrix. The updated state and covariance matrix are calculated as:(16)$$\begin{array}{cc} x_{t} & ={\hat{x}}_{t}+K_{t}\left(z_{t}-h\left({\hat{x}}_{t}\right)\right), \end{array}$$(17)$$\begin{array}{cc} P_{t} & =\left(I-K_{t}H_{t}\right){\hat{P}}_{t}, \end{array}$$
where the innovation residual vector ($z_{t}-h\left({\hat{x}}_{t}\right)$) contains the the measurements model vector ($h({\hat{x}}_{t})$) and observation vector ($z_{t}$). When the epoch contains GNSS measurements, then the step after the filter update is the integer ambiguity resolution on all of the baselines. Least-squares AMBiguity Decorrelation Adjustment [18] method is applied for the integer ambiguity resolution on the baseline between the permanent and the moving base receivers. The moving baseline-based estimation allows to extend the optimization with an attitude-related constraint [19,20,21,22]. The attitude estimation in this approach is based on the quaternion representation, accordingly, the constraint is the norm of the quaternions, as in a previous work [16]. The use of simultaneous processing of the moving baseline further increases the reliability of the estimation algorithm and reduces the convergence time of sensor bias states.

## 4. Cycle Slip Detection Techniques for Moving Platforms

The GNSS carrier-phase observations are sensitive to the successful resolution of integer phase ambiguities. Thus, cycle slips must be detected to re-initialize the estimation of the integer ambiguities in the state vector. An ignored cycle slip corrupts the estimated integer ambiguity of the given satellite, thus, the usage of an incorrect value can lead to degradation of estimation accuracy (Figure 3).

We will introduce the studied cycle slip detection methods in the following subsections and introduce our approach utilizing the information on the kinematics of the moving platform provided by the EKF prediction.

### 4.1. Loss-of-Lock Indicator-Based Cycle Slip Detection

A commonly used technique to detect cycle slips is to use the LLI information provided by the receiver. Receivers continuously monitor the availability of the satellite signals in the consecutive epochs during tracking. When a signal outage occurs, a Loss-of-Lock Indicator is stored in the observation files. In case of loss-of-lock, the phase ambiguities needs to be re-initialized, thus, falsely fixed ambiguities do not deteriorate the positioning accuracy. The advantage of this method is that the LLIs are logged in the receiver, thus, no further data processing is needed to issue a new ambiguity parameter for the respective satellite. However, cycle slips can occur even when the observations are available in the consecutive epochs. This is especially true under poor satellite visibility conditions: small number of satellites and significant obstructions causing frequent signal losses. Thus, the LLI method is unable to detect all of the cycle slips, leading to false ambiguity resolution and accuracy deterioration. Thus, more sophisticated methods need to be used for cycle slip detection.

### 4.2. Cycle Slip Detection Using the Melbourne–Wübbena Linear Combination

The theory of cycle slip detection using dual-frequency GPS observations was developed several decades ago for geodetic receivers. Recently, several low-cost GNSS chipset manufacturers have released their multi-frequency receivers. Thus, the dual-frequency cycle slip detection methods can also be applied for these receivers. One of the techniques is based on the Melbourne–Wübbena linear combination of dual frequency pseudorange and phase observations. The idea is to form the wide-lane linear combination of carrier-phase observations and compare it to the narrow-lane linear combination of the pseudorange measurement
(18)$$\begin{array}{c} \phi_{MW}=\phi_{WL}-\varrho_{NL}=\frac{f_{m}\phi_{m}-f_{n}\phi_{n}}{f_{m}-f_{n}}-\frac{f_{m}\varrho_{m}+f_{n}\varrho_{n}}{f_{m}+f_{n}}=\lambda_{WL}N_{WL}, \end{array}$$
where $f_{m}$ and $f_{n}$ are the carrier frequencies of the used measurements, $\lambda_{WL}$ is the wide-lane wavelength ($\lambda_{WL}=c/\left(f_{m}-f_{n}\right)$) and $N_{WL}$ is the wide-lane ambiguity. The wide-lane combination of carrier-phases has the wavelength of ca. 86 cm, thus, it enlarges the ambiguity spacing. Subtracting the narrow-lane linear combination of the pseudoranges, one can calculate the WL ambiguity. However, it must be mentioned that the observation error of the pseudoranges can significantly increase in the case of high multipath errors or in the case of low-cost instruments, such as dual-frequency cell phones [6].

### 4.3. The TurboEdit Cycle Slip Detection Method

The TurboEdit method was first introduced by Blewitt [5] and it aims at detecting cycle slips even when the pseudorange observations have lower accuracy. It introduces the phase ionospheric residual (PIR) combination to the cycle slip detection algorithm,
(19)$$\begin{array}{c} \phi_{PIR}=\phi_{m}-\phi_{n}=\lambda_{m}N_{m}-\lambda nN_{n}+\frac{f_{m}^{2}-f_{n}^{2}}{f_{n}^{2}}I. \end{array}$$ The method uses the moving-window averaging filtered MW and PIR combination values for detecting cycle slips. The MW and the PIR extended TurboEdit methods rely heavily on the continuity of the multi-frequency measurement data and low measurement noise, which cannot be guaranteed in urban, rapidly changing environments. In the next section, we introduce a different approach for cycle slip detection that leverages the fusion of other navigation sensors with GNSS.

### 4.4. The Prediction-Based Cycle Slip Detection Algorithm

Since GNSS is subject to signal losses in urban canyons, the localization of autonomous vehicles is achieved by the fusion of several navigation sensors. We propose a GNSS cycle slip detection algorithm that uses not only GNSS observations, but the observations of other navigation sensors, too. The proposed cycle slip detection method is based on the triple differences of the carrier-phase measurements and the predicted states of the EKF algorithm. In geodetic positioning, triple differences are widely used to detect and even repair cycle slips [23]. Although this method works well in static measurement where the reduction of the hardware and atmospheric delays by the differentiation is sufficient for the cycle slip detection, the dynamic effects of the moving receivers cannot be compensated in this way. The presented method corrects the aforementioned deficiency by the dynamic information from the estimation algorithm. The predicted triple-differenced carrier-phase measurements (Equation 20) are calculated based on carrier-phase sensor model (Equation 2) using the calculated satellite positions, clock biases, relativistic, group and atmospheric delays and the estimated position, attitude, receiver clock biases, the inter-channel biases and the integer ambiguity states from the EKF prediction in the actual and the previous GNSS measurement epochs (${\hat{x}}_{t_{GNSS}},{\hat{x}}_{t_{GNSS}-1}$).
$$\begin{array}{cc} \; & \phi_{rb}^{jk}{|}_{t_{GNSS}-1}^{t_{GNSS}}\left({\hat{x}}_{t_{GNSS}},{\hat{x}}_{t_{GNSS}-1}\right)=\\ & (\left(\phi_{r}^{j}\left({\hat{x}}_{t_{GNSS}}\right)-\phi_{b}^{j}\left({\hat{x}}_{t_{GNSS}}\right)\right)-\left(\phi_{r}^{k}\left({\hat{x}}_{t_{GNSS}}\right)-\phi_{b}^{k}\left({\hat{x}}_{t_{GNSS}}\right)\right))- \end{array}$$
(20)$$\;\;\;\;\;\;\;\;\;\;\;\;\;\;\;\;\;\begin{array}{c} \left(\left(\phi_{r}^{j}\left({\hat{x}}_{t_{GNSS}-1}\right)-\phi_{b}^{j}\left({\hat{x}}_{t_{GNSS}-1}\right)\right)-\left(\phi_{r}^{k}\left({\hat{x}}_{t_{GNSS}-1}\right)-\phi_{b}^{k}\left({\hat{x}}_{t_{GNSS}-1}\right)\right)\right) \end{array}$$

The triple-differenced values are also calculated from the raw GNSS measurements ($\phi_{rb}^{jk}{|}_{t_{GNSS}-1}^{t_{GNSS}}\left(\mathrm{GNSS}\mathrm{rec}\right)$).
(21)$$\begin{array}{c} |\phi_{rb}^{jk}{|}_{t_{GNSS}-1}^{t_{GNSS}}\left({\hat{x}}_{t_{GNSS}},{\hat{x}}_{t_{GNSS}-1}\right)-\phi_{rb}^{jk}{|}_{t_{GNSS}-1}^{t_{GNSS}}\left({\mathrm{GNSS}}_{\mathrm{rec}}\right)|>\mathrm{threshold} \end{array}$$ In case the absolute difference between the predicted and the receiver data-derived triple differences is greater than a predefined threshold (Equation 21), a cycle slip is reported in the algorithm. In this study, the threshold is set to half a cycle as a constant, which is a commonly used value [9,24]. Using the error propagation and the EKF covariance matrix, it is possible to estimate the uncertainty of the predicted triple differenced carrier-phase measurements. The realization of this calculation is an opportunity for future development. The detection algorithm defines the entry of a satellite into a measurement as an ambiguity state initialization. In the case of continuous satellite reception, the dynamical compensation works effectively within a short interval. In the case of longer outage, the algorithm defines cycle jump, so the slowly changing inertial sensor drift has less effect in the detection. The inertial bias states are updated by the tightly coupled sensor fusion during continuous GNSS data receiving. The cycle slip detection is realised in the ambiguity space. Using the GNSS measurement models to predict the TD carrier-phase values eliminates the further projections between ambiguity the position or the attitude space. The proposed method can be applied to single-frequency measurements and moving baselines too. Our approach is validated on three test scenarios using real observations obtained in open-sky and urban canyon environment. The measurement platform and the case studies are discussed in the following sections.

## 5. Measurement Platform

A ground vehicle platform was chosen for testing the various cycle slip detection techniques in different case studies.

Figure 4 and Table 1 depict the applied setup. The antenna of the moving base receiver (UBX A1) is the centre of the body coordinate system. The x-axis points forward, the y-axis points to the right and the z-axis points down. The u-blox antennas were placed in a triangular shape and the distances were maximized to be sensitive to all three attitude angles.

We used three independent u-blox ZED-F9P low-cost multi GNSS, multi-frequency receivers with ANN-MB series patch antennas. These receivers provided raw, 10 Hz frequency GPS, GLONASS, Galileo, Beidou pseudorange, carrier-phase and Doppler measurements for post-processing. The accelerometer, angular velocity, magnetic field and barometric data were collected by a PixFalcon Flight Control Computer (INS data 50 Hz; magnetic, barometric data 10 Hz measurement frequency). A tactical grade KVH GEO-FOG 3D Dual unit was used to collect reference data (100 Hz) for comparisons. This unit is equipped with a dual-frequency Trimble MB-Two GNSS lightweight receiver capable of providing position and heading information with 8 mm and 0.01 degrees accuracy, respectively. This KVH unit contains a Microelectromechanical Systems (MEMS) accelerometer as well as a high-accuracy Fiber Optic Gyro (FOG).

## 6. Case Study 1

A measurement campaign was organized in 2020 at the ZalaZone automotive test track [25]. During this campaign, we had the opportunity to collect measurement data with an ideal sky-view and high horizontal dynamics. The reference GNSSNet.hu-ZALA GNSS base station was tracking the GPS and GLONASS systems on L1 and L2 with the measurement rate of 1 Hz. The distance between the base station and the moving base was approximately 6 km. The RTCM stream was provided to the KVH unit and to the u-blox receivers in real time.

We drove our vehicle in the Smart City Zone, where we could record data with high dynamic cornering, braking and several figure-eight maneuvers, the trajectory of the drive is depicted in Figure 5 and the total measurement time was 870 s. The integer ambiguity resolution (IAR) success rates of the different solutions on the baselines are depicted in Table 2. The KVH unit resolved the integer ambiguities in 100% of the epochs, the moving base u-blox receiver (UBX) did it in the 88.8% of the time on the permanent base-moving base baseline ($Bl_{1}$). The IAR rate value of the Melbourne–Wübbena (MW) method-based detection was 99.2%, and the TurboEdit (TE) method reported successful ambiguity resolution in the 99.87% of the epochs. The EKF algorithm uses GNSS processing of the moving baseline for simultaneous attitude estimation. The IAR ratios on the moving baselines ($Bl_{2}$, $Bl_{3}$) are 100%, except the TE method which was reported as 96.77%.

The KVH unit was selected as position reference in this case. Figure 6 depicts the horizontal, the vertical coordinates the KVH unit and the TC algorithm based on the PBCS cycle slip detection method.

Table 3 contains the mean, the standard deviation and the maximum absolute values of the lever arm-compensated navigational coordinate differences. The lever arm compensation is based on the KVH attitude angles and the lever arm between the KVH sensor and the moving base receiver. The Fix. solution flag denotes the commonly fixed epochs on the permanent base–moving base baseline. The Flo. flag means that one of the compared solutions did not give a successful integer ambiguity resolution during the measurement epoch. The mean, the standard deviation and maximum absolute position differences from the reference are used to determine the accuracy of the given solution.

Significant differences cannot be detected in the coordinate statistics (Table 3) although the imperfection of the lever arm compensation incorporates a larger maximum value in all of the coordinates. The horizontal and vertical velocity statistics show a high degree of consistency (Table 4).

The calculated attitude angles (ϕ—roll, θ—pitch, ψ—yaw) during the measurement and the differences between the TC PBCS and the reference KVH unit are depicted in Figure 7.

The non-significant differences in statistical data of the attitude angle comparison (Table 5) confirm the high IAR ratios on the moving baselines and the accuracy of attitude estimation.

Processing the validation measurement showed that the PBCS cycle slip method can achieve high accuracy with high reliability in an open-sky environment.

## 7. Case Study 2

The next step after the successfully validation of the proposed PBCS detection method is the testing in the more challenging urban canyon environment. Case Study 2 uses the same measurement platform and sensors as the validation measurement. The car was driven around ten-storey buildings with two sections of more open areas at the start and during the middle of the path (Figure 8). This environment causes significant multipath error in measurements, the masking effect of the buildings is also high. In this case, the 30 m building height and 15 m horizontal distance from the road limits the elevation angle of the visible satellites to 60–90 degrees. The duration of the test drive was 350 s.

The permanent base receiver was the BUTE EUREF Permanent Station, which provided real-time GPS/GLONASS and Galileo corrections for RTK positioning and was approximately 4.5 km. In this case, the KVH unit was operating in absolute positioning mode. Figure 9 depicts the number of all available moving base receivers’ raw carrier-phase measurement for L1 only and L1, L2 frequencies together. The intervals with a low number of available satellites (2–5) due to the blockage caused by the urban canyon are clearly visible.

The integer ambiguity success rates are summarized by Table 6. The success rate of the KVH was 0% due to the absolute positioning mode, the UBX solution was reported at 92.96% rate. The LLI-based TC algorithm obtained the highest success rate (79.08%), while the PBCS method had 75.34% on $Bl_{1}$. The linear combination-based MW and TE solutions perform much worse in this scenario. The absence of a continuous dual frequency GNSS data makes convergence of the moving average filter difficult. This can also be observed on the moving baseline IAR rates, while LLI-based and the PBCS-based solutions achieve more than 90% percent.

In this case, the UBX solution was chosen as the position reference due the RTK positioning. The horizontal, the vertical position and the differences from the reference are depicted in Figure 10.

These results are summarized by Table 7. The position difference statistics show no significant deviances between the PBCS and LLI methods in either float or fixed cases.

The velocity statistics (Table 8) confirm the struggle of the linear combination-based cycle slip methods. The higher horizontal, vertical standard deviations and maximum absolute values of the velocities indicate higher jumps in the position solution.

Although the KVH unit was operating in absolute positioning mode, the fusion of the FOG-based INS and the dual antenna heading system provided a highly accurate attitude estimation. This data are the chosen attitude reference in the comparison. The attitude angles of the KVH unit and the PBCS-based TC solution and the differences between them are depicted in Figure 11.

During a cruise of the urban canyon area, a jump is detectable in the pitch and yaw angles around the 250th second of the measurement. This was caused by a false resolved integer ambiguity which distorted the mentioned angles. The effect of this jump was reduced by the filter in 15 s.

These jumps are also detectable in the maximum absolute differences at PBCS and LLI methods in Table 9. The mean, standard deviation and the maximum values do not show significant differences between PBCS and LLI methods. The TE and MW-based estimations show a high degree of deviation in all of the attitude angles, which is also confirmed by the ambiguity resolution rates on the moving baselines.

## 8. Case Study 3

The measurement platform and the location of the measurement was the same as in Case Study 2. The permanent base was the BUTE EUREF Permanent Station again, and the trajectory was also the same. In this case, the KVH GEO-FOG 3D Dual was also working in RTK mode.

Figure 12 shows the number of tracked satellites providing dual (L1 + L2) and single (L1) carries phase measurements. This diagram highlights the difficulties of the measurement scenario. The number of available satellites often decreases under five at the more difficult sections and there are also intervals without carrier-phase measurements. This is the key difference between this test and Case Study 2, where the number of satellites with available L1 and L1+L2 carrier-phase measurements was greater.

The measurement results include the internal position solution of the moving base u-blox receiver (UBX), the KVH GEO-FOG 3D Dual unit’s position and attitude solutions (KVH), and the TC algorithm result using the PBCS, the LLI, the MW and the TE-based cycle slip detection methods. Our intention was to use the position and attitude solution of the KVH GEO-FOG 3D Dual as references for this experiment, since this tactical grade unit is equipped with high-end sensors. However, the results of the KVH unit show significant differences compared with the UBX, PBCS, LLI, solutions. The first major difference is in the ratio of epochs with successful ambiguity resolution (Table 10). The internal solution of the moving base u-blox receiver reported 79.76% success rate. The high ambiguity resolution success rate and the RTK performance of the u-blox ZED-F9P receiver was also presented by Takasu [26]. The ambiguity success rate of the TC LLI solution was 56.31% on the permanent base-moving base baseline ($Bl_{1}$), compared to that the PBCS method increases the success rate by nearly 19% (to 75.44%). Surprisingly, the tactical grade KVH unit had successful ambiguity resolution only in 48.24% of the epochs, hence, it is not straightforward to use its solution as a reference in this case. The MW and the TurboEdit-based processing results show similar success rates. The TC algorithm tries to resolve the integer ambiguities on the moving baselines ($B_{2}$, $Bl_{3}$) using the quaternion constraint LAMBDA method. The PBCS-based TC processing shows the highest success rates on the moving baselines (77.96%, 79.56%), they are 10% better than LLI based estimation. The linear combination-based TE method shows a slightly higher than 40% success rate. The MW cycle slip detection-based estimation shows the lowest (20.47% and 10.62%) rates. This can be explained by the fact that the MW method relies heavily on the pseudorange measurements in the high multipath environment. However, the success rate does not give a clear conclusion of the quality of the results, especially in such challenging environment.

One should also examine the roughness of the calculated trajectory to assess the performance of the different solutions. High coordinate jumps can be seen on the trajectory solution of the u-blox receiver (UBX) in the urban canyon section. The receiver even switched to absolute 3D positioning from RTK mode in some cases. The UBX solution lacking the inertial aiding shows position jumps of several meters in horizontal and nearly 50 m in vertical direction. Figure 13 depicts the trajectory, the vertical changes and the differences from the UBX solution.

The second contradiction of the KVH unit can be seen in Figure 14. The coordinate solution seems purely inertial sensor-based in the low satellite visibility sections. Neither of the high accuracy inertial sensors can handle the long outage, the solution is drifting in the absence of GNSS data. In contrast, the u-blox receivers and the TC algorithms report mostly float RTK solutions. When GNSS reception of the KVH sensor is improved, the receiver reports fixed RTK position but the solution is discontinuous. The highest jump in absolute fixed coordinates of the KVH unit are several meters horizontally and 16 m vertically (Table 11). The false fixed positions by the KVH unit are also depicted in Figure 14, using a satellite view of the area as background.

The statistics of navigational coordinate system position differences from the reference UBX solution are depicted in Table 11. The PBCS method-based solution shows the lowest differences in the commonly fixed epochs with the highest ambiguity resolution success rate. The results of LLI method-based solution show higher standard deviation and maximum values for the commonly fixed epoch caused by a false fixed interval in the estimation.

Although occasionally all solutions suffer from false resolved integer ambiguities (maximum absolute differences rows in Table 11), the proposed cycle slip detection method performs better and has largely smoothest solutions. This is confirmed by horizontal and vertical velocities derived from coordinate differences which are used to quantify the continuity of the position estimation (Table 12).

Among the sensor fusion methods, the KVH solution shows the highest standard deviation and maximum values both horizontally and vertically. This is caused by high coordinate jumps during the re-initialization after the drifted sections. The u-blox internal (UBX) estimation also suffers from high coordinate jumps, but this can be explained by the fact that this sensor is not aided by any inertial data. The PBCS-based solution shows the lowest standard deviation of the both the horizontal and vertical instantaneous velocities. The low standard deviation and maximum values confirm the reliability of the position estimation shown in Figure 14. The MW and the TE show slightly larger standard deviation values, while their maximums are significantly higher than both the PBCS and the LLI methods. A possible explanation is that these techniques rely heavily on the continuous availability of accurate pseudorange observations.

The high positioning error precluded the use of the KVH sensor as a reference in case of the position estimation. However, the FOG-based attitude estimation had high accuracy during the test and it is used as reference to the comparison of the Euler angles.

Figure 15 depicts the attitude angles (roll—ϕ, pitch—θ, yaw—ψ) and the differences from the KVH unit. The heading angle is drifting only in the very weak satellite signal reception sections. Attitude angle jumps can be observed at the float-to-fixed transitions on the moving baselines, when the resolved integer ambiguities correct the attitude estimation. The attitude angle differences are summarized in Table 13.

The PBCS, LLI and the TE-based processing results show low mean and standard deviation difference values compared to the reference data, the absolute values are lower than 0.6 degree. The MW method-based solution shows higher statistical differences in the attitude estimation. This is related to the low integer ambiguity success rates on the moving baselines, while the corresponding attitude estimation suffers from inadequate cycle slip detection on the moving baseline.

## 9. Discussion

The present paper proposed a prediction-based cycle slip detection method, which is based on a tightly coupled sensor integration method of multi-constellation, multi-frequency, multi-baseline GNSS measurements and inertial, magnetic, barometric measurements. The main novelty of the proposed approach is to use the predicted states of the navigation EKF in every single epoch to handle the possible cycle slips at position and attitude estimating baselines to improve the ambiguity fix ratio. The estimated states are used to model the dynamic evolution of the receiver-satellite-time-differentiated (triple differenced) carrier-phase measurements and by a comparison of them to the values derived from raw GNSS measurements, cycle slips and outliers in pseudorange measurements are detected. The aim of the method is to reduce the atmospheric, hardware delays through differentiating, and the effects of movement through dynamic compensation, which is not possible with simple differencing. The method was validated in an open-sky environment first, where the position data and the moving baseline based attitudes were estimated with high accuracy compared with a tactical grade reference system. The superiority of the prediction-based cycle slip detection method in tightly coupled estimation algorithm was proven in urban canyon tests. The proposed method-based estimation was shown in a challenging environment (Case Study 3). It gave the highest ambiguity rates, the least amount of position jumps, the lowest coordinate attitude angle differences from the references compared to the simple loss-of-lock indicator based and the more sophisticated linear combination-based Melbourne–Wübbena and TurboEdit methods. Furthermore, the low-cost tightly coupled integration algorithms also performed better at this measurement scenario than the tactical grade KVH GEO-FOG 3D Dual fiber optic gyro-based navigational unit that was originally intended to be used as a validation reference for urban cases as well. The presented methods are working in post-processed mode, but their real-time implementation is currently under study as the next step for their adoption in autonomous test vehicles operating in cities.

## Figures and Tables

**Figure 1 sensors-23-02141-f001:**
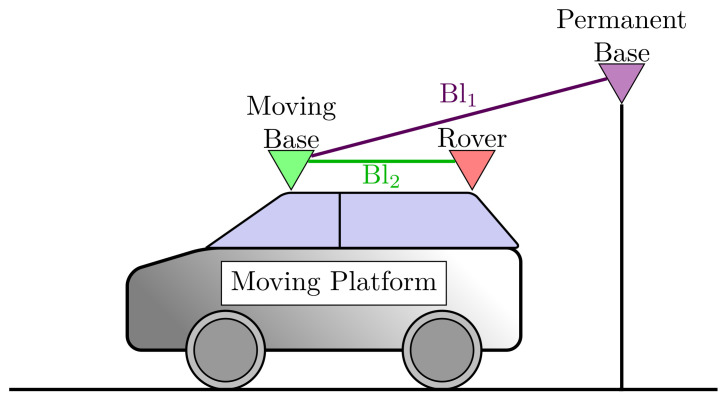
Receiver types in the multiple baseline model.

**Figure 2 sensors-23-02141-f002:**
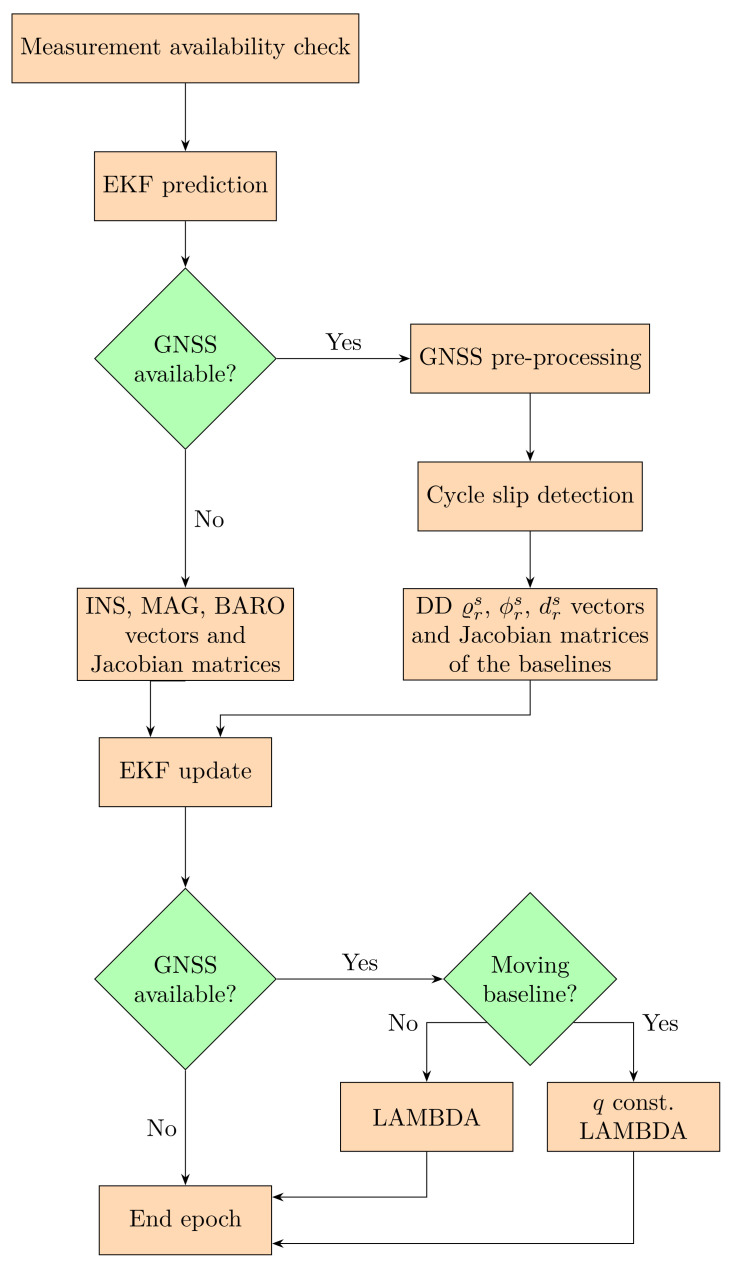
Flowchart of the EKF-based estimation algorithm in a single measurement epoch.

**Figure 3 sensors-23-02141-f003:**
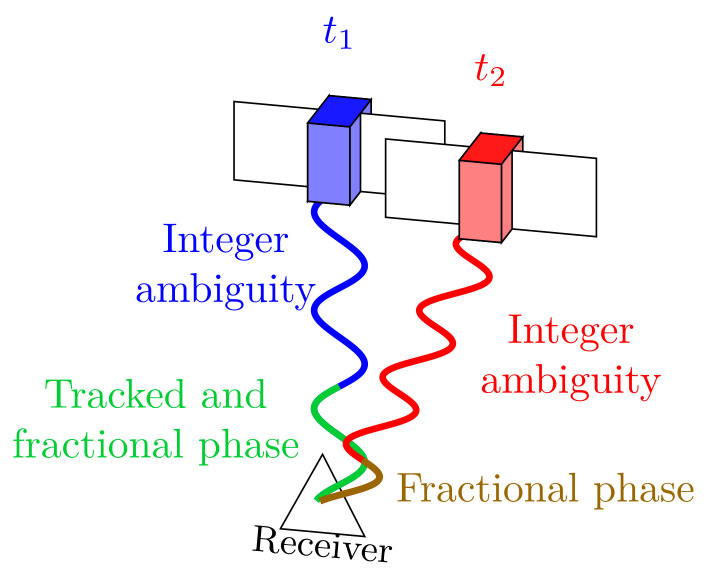
Carrier-phase measurement during cycle slip.

**Figure 4 sensors-23-02141-f004:**
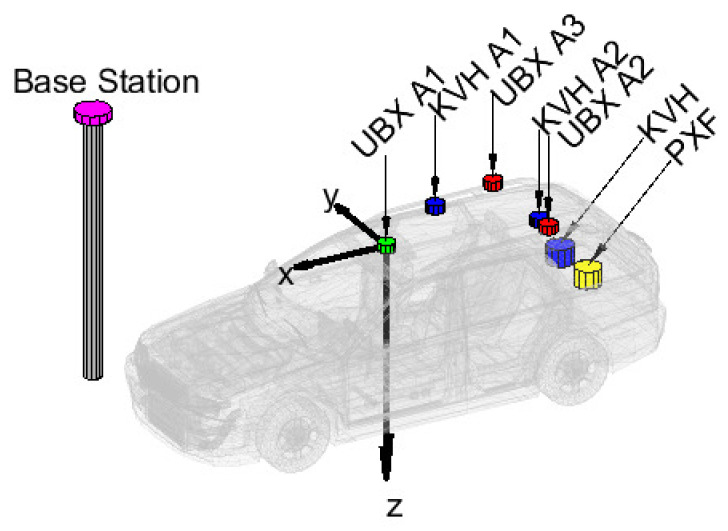
Measurement setup.

**Figure 5 sensors-23-02141-f005:**
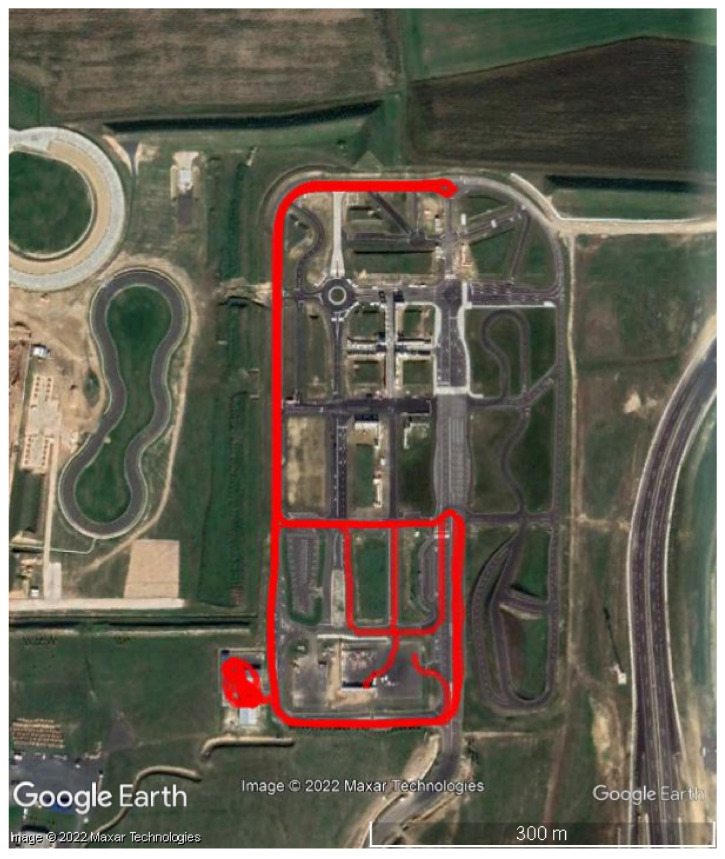
Trajectory of Case Study 1.

**Figure 6 sensors-23-02141-f006:**
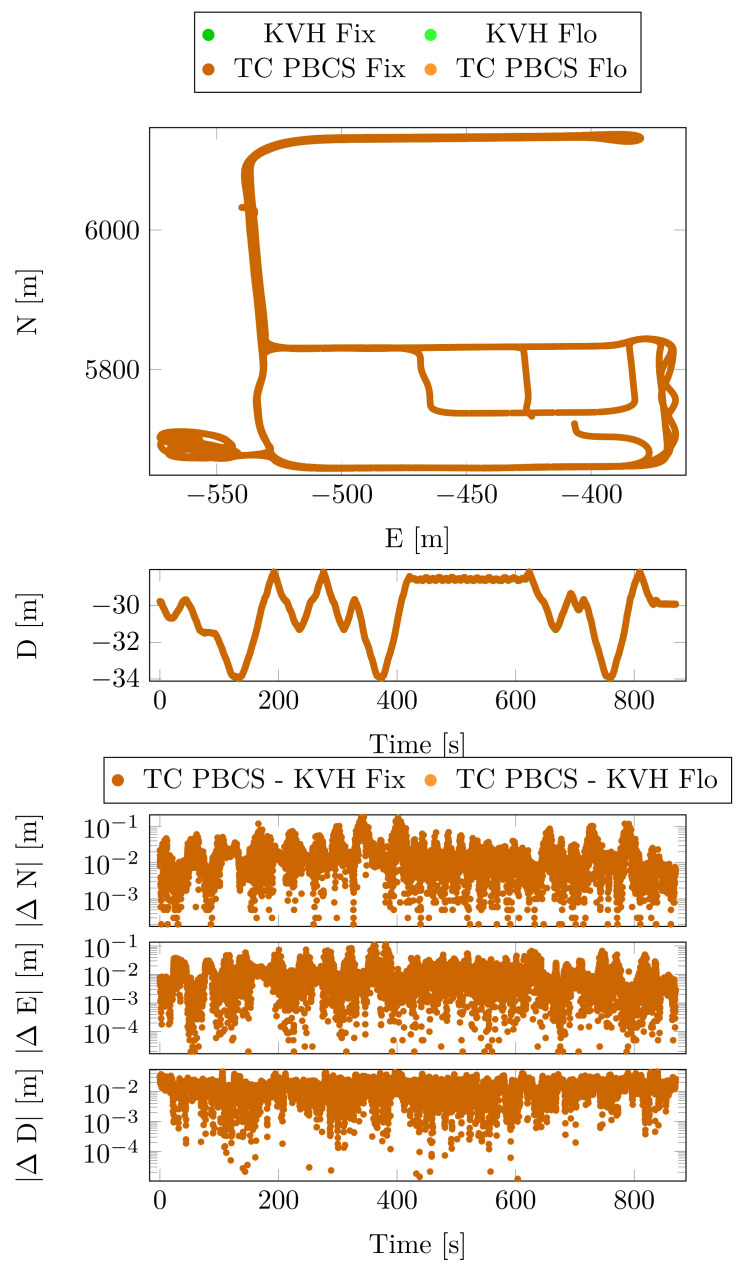
Position and position differences from reference—Case Study 1.

**Figure 7 sensors-23-02141-f007:**
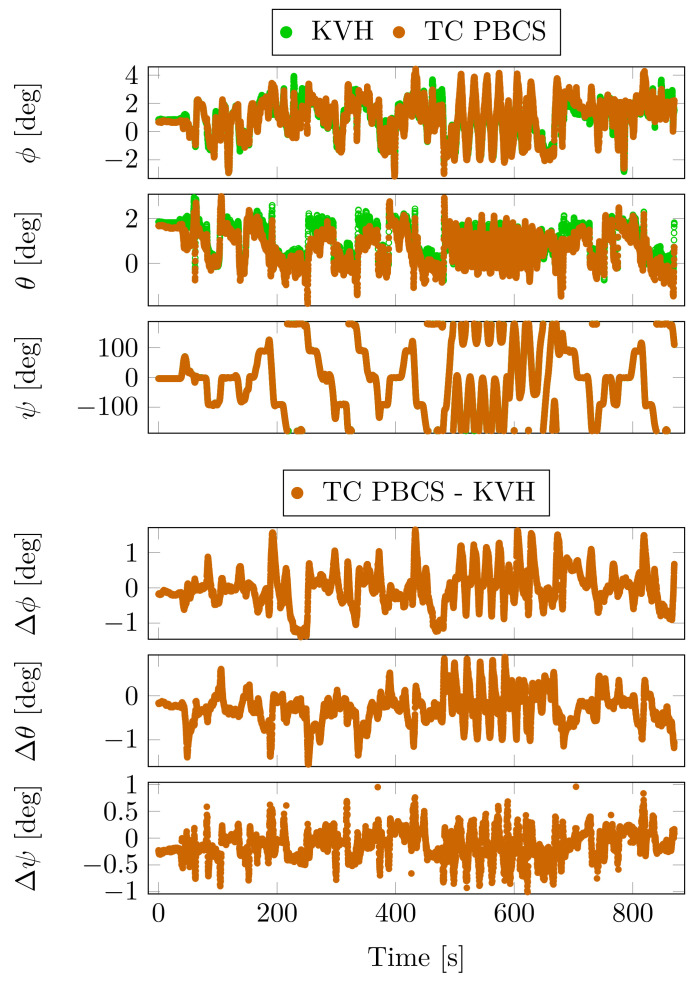
Attitude angles and differences from the reference—Case Study 1.

**Figure 8 sensors-23-02141-f008:**
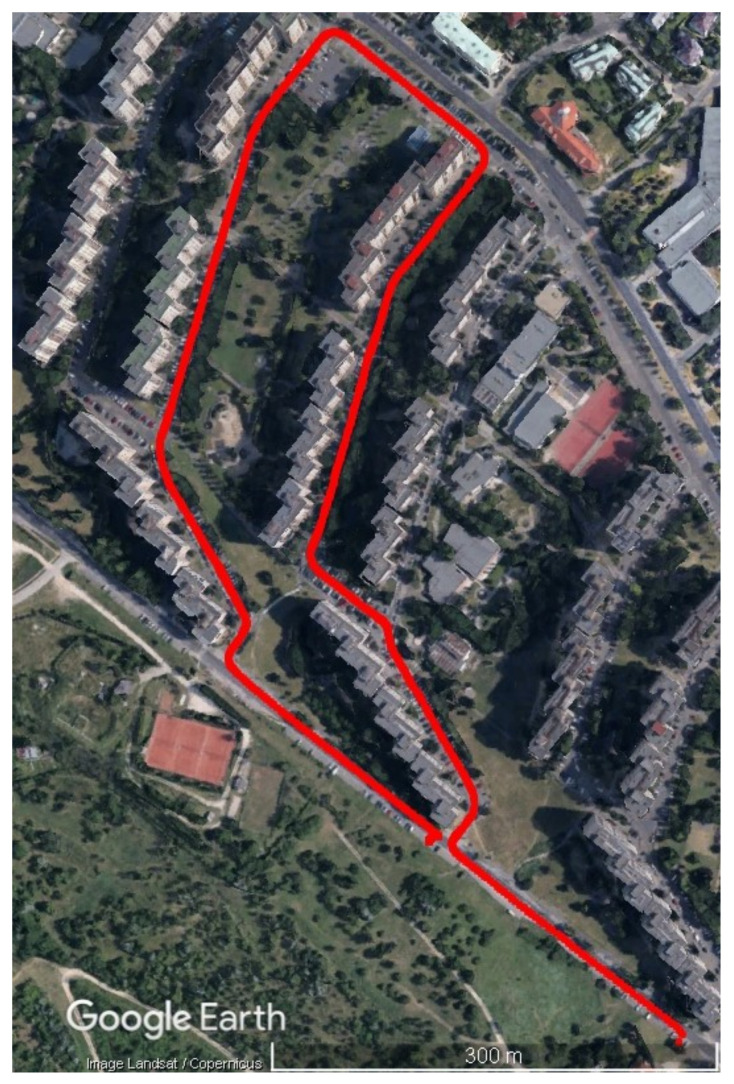
Trajectory of the urban canyon test.

**Figure 9 sensors-23-02141-f009:**
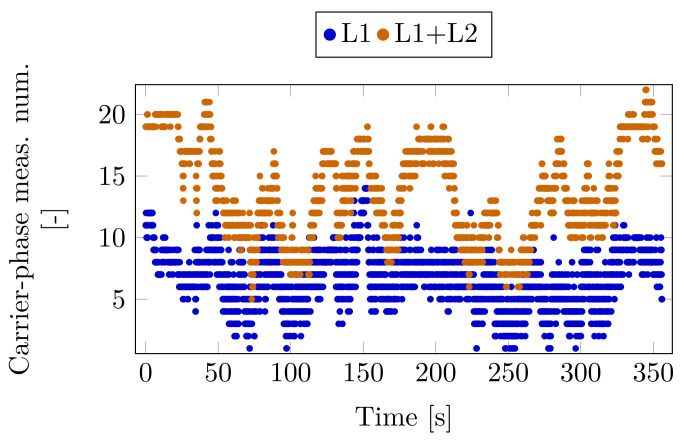
Carrier-phase measurement number—Case Study 2.

**Figure 10 sensors-23-02141-f010:**
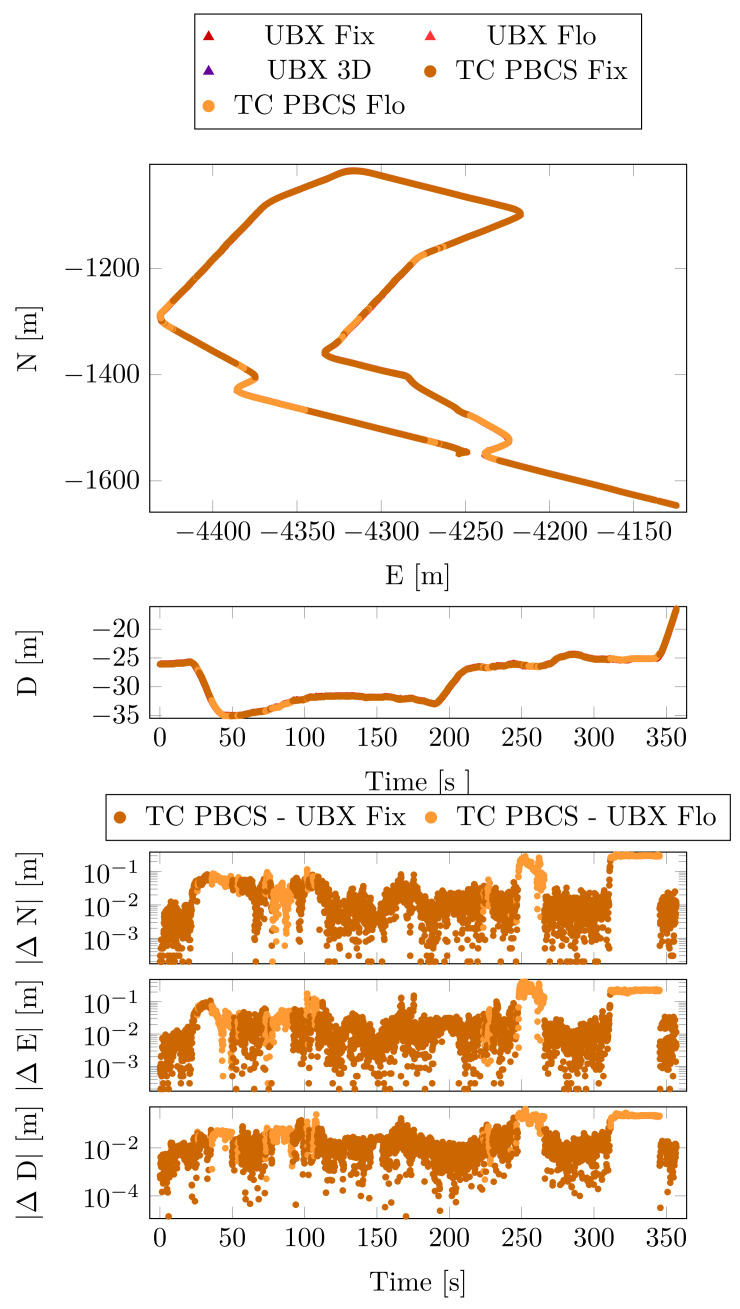
Position and position differences from the reference—Case Study 2.

**Figure 11 sensors-23-02141-f011:**
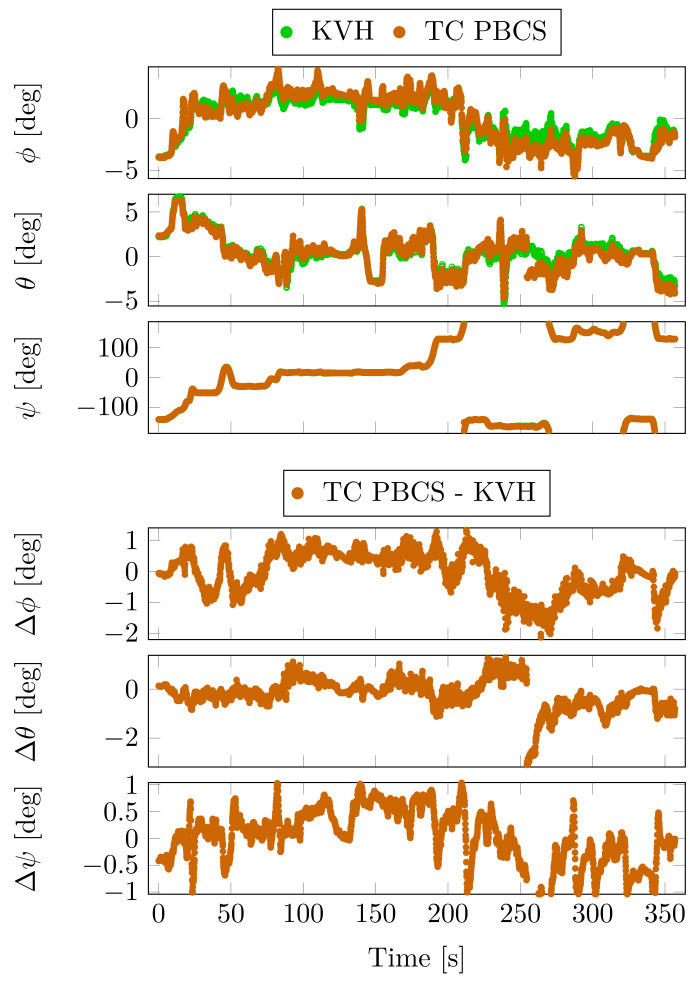
Attitude and attitude angle differences from KVH unit—Case Study 2.

**Figure 12 sensors-23-02141-f012:**
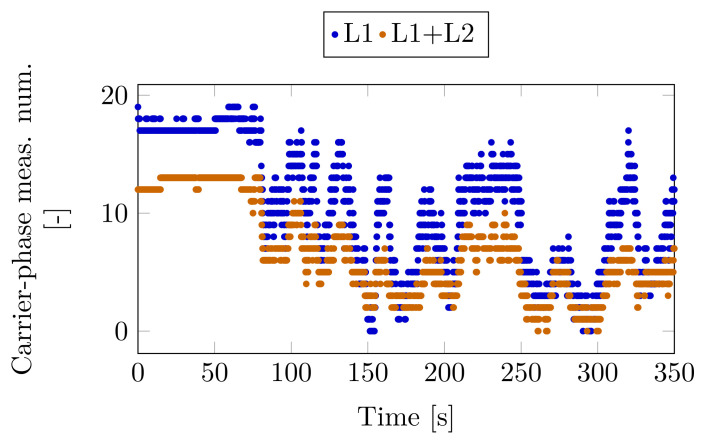
Number of carrier-phase measurements on the two frequencies together and on one frequency only—Case Study 3.

**Figure 13 sensors-23-02141-f013:**
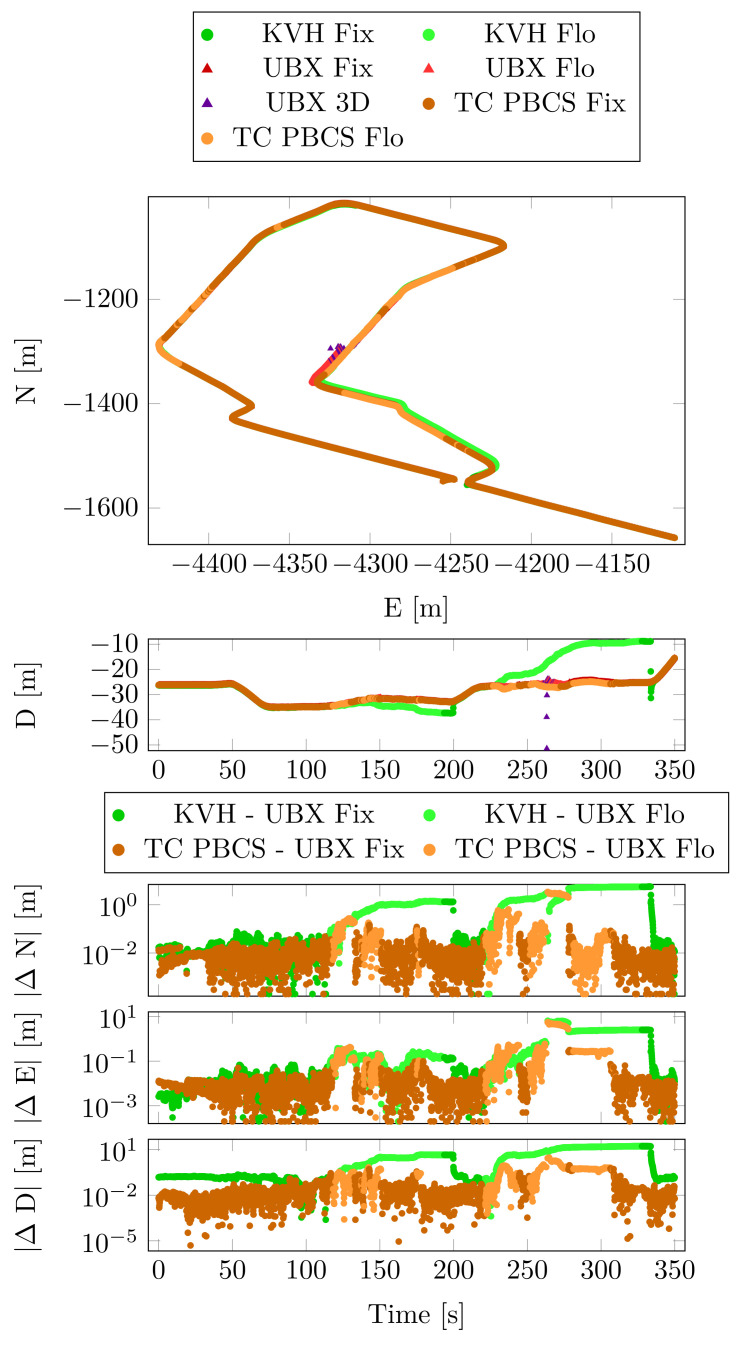
Position and position differences from the reference—Case Study 3.

**Figure 14 sensors-23-02141-f014:**
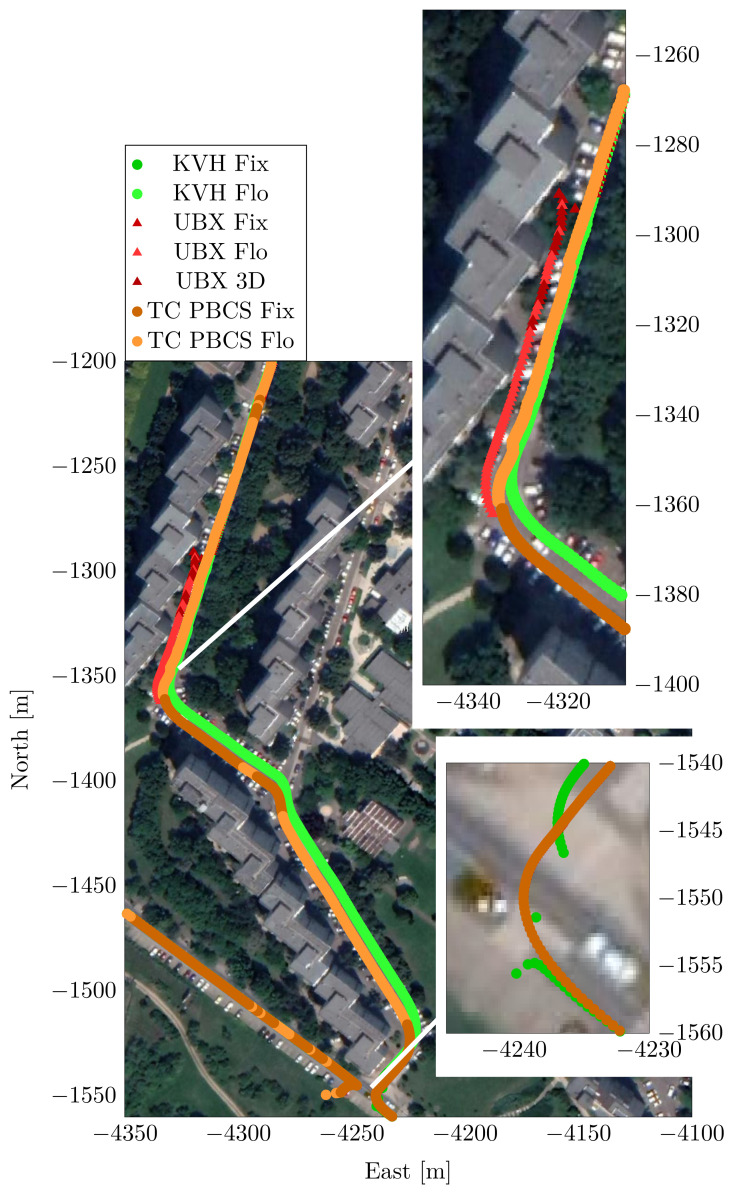
Trajectory in the deep urban canyon environment—Case Study 3.

**Figure 15 sensors-23-02141-f015:**
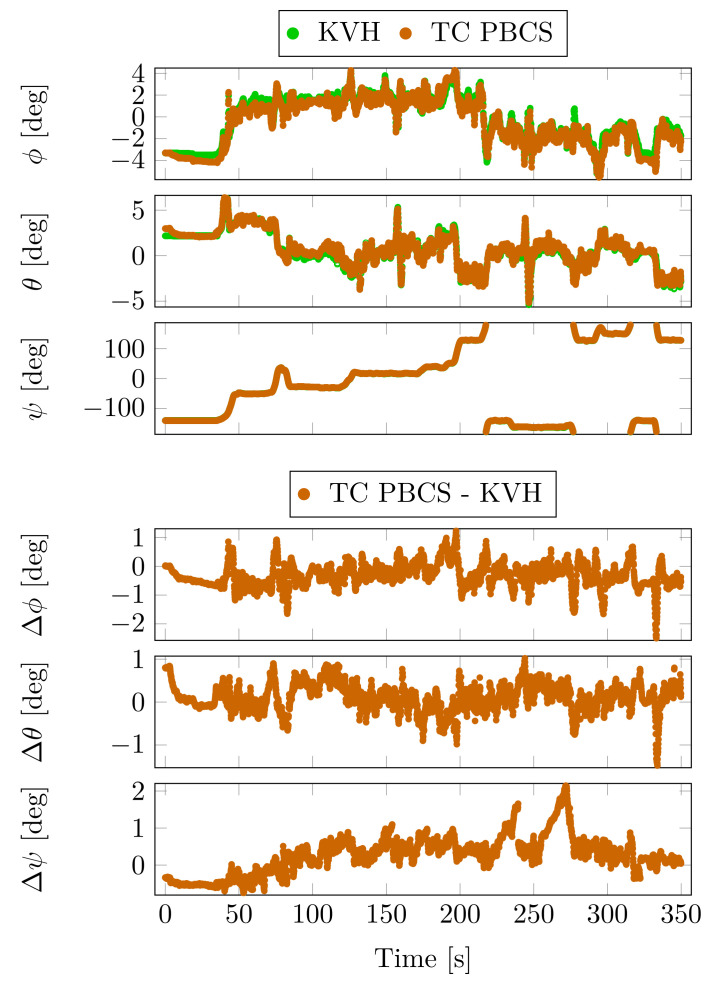
Attitude and attitude angle differences from the KVH unit—Case Study 3.

**Table 1 sensors-23-02141-t001:** Lever arms between the applied sensors in meters.

	UBX A1	UBX A2	UBX A3	PXF	KVH	KVH A1	KVH A2
x [m]	0.00	−0.98	−0.98	−1.80	−1.80	−0.40	−0.98
y [m]	0.00	−0.45	0.45	0.00	0.29	0.37	−0.26
z [m]	0.00	−0.08	−0.08	0.90	0.90	−0.03	−0.03

**Table 2 sensors-23-02141-t002:** Ambiguity resolution success ratios of the different solutions—Case Study 1.

	PBCS	LLI	MW	TE	UBX	KVX
*Bl*_1_ AR [%]	100.00	99.19	99.20	99.87	88.80	100.0
*Bl*_2_ AR [%]	100.00	100.00	100.00	96.77	-	-
*Bl*_3_ AR [%]	100.00	100.00	100.00	96.77	-	-

**Table 3 sensors-23-02141-t003:** Coordinate difference statistics from KVH GEO-FOG 3D—Case Study 1.

	*N_PBCS_*	*N_LLI_*	*N_MW_*	*N_TE_*	*N_UBX_*
Fix. Mean [m]	0.01	0.01	0.01	0.01	0.00
Fix. Std. [m]	0.02	0.02	0.02	0.02	0.01
Fix. Max(abs) [m]	0.18	0.20	0.20	0.25	0.08
Flo. Mean [m]	0.00	−0.50	−0.52	0.07	0.03
Flo. Std. [m]	0.00	0.02	0.02	0.01	0.25
Flo. Max(abs) [m]	0.00	0.53	0.54	0.09	0.67
	*E_PBCS_*	*E_LLI_*	*E_MW_*	*E_TE_*	*E_UBX_*
Fix. Mean [m]	0.01	0.01	0.01	0.01	0.01
Fix. Std. [m]	0.01	0.01	0.01	0.01	0.01
Fix. Max(abs) [m]	0.11	0.11	0.11	0.22	0.05
Flo. Mean [m]	0.00	0.42	0.43	0.12	0.15
Flo. Std. [m]	0.00	0.01	0.01	0.08	0.11
Flo. Max(abs) [m]	0.00	0.44	0.45	0.22	0.41
	*D_PBCS_*	*D_LLI_*	*D_MW_*	*D_TE_*	*D_UBX_*
Fix. Mean [m]	0.01	0.01	0.01	0.01	0.00
Fix. Std. [m]	0.01	0.01	0.01	0.01	0.01
Fix. Max(abs) [m]	0.05	0.07	0.07	0.10	0.04
Flo. Mean [m]	0.00	−0.99	−1.05	0.10	−0.20
Flo. Std. [m]	0.00	0.04	0.04	0.04	0.43
Flo. Max(abs) [m]	0.00	1.08	1.14	0.13	1.12

**Table 4 sensors-23-02141-t004:** Horizontal and vertical velocity statistics—Case Study 1.

	*H_PBCS_*	*H_LLI_*	*H_MW_*	*H_TE_*	*H_UBX_*	*H_KVH_*
Mean [m/s]	7.31	7.31	7.31	7.32	7.31	7.31
Std [m/s]	4.14	4.14	4.14	4.15	4.14	4.14
Max(abs) [m/s]	21.94	21.94	21.94	22.22	21.58	21.58
	*V_PBCS_*	*V_LLI_*	*V_MW_*	*V_TE_*	*V_UBX_*	*V_KVH_*
Mean [m/s]	0.00	0.00	0.00	0.00	0.00	0.00
Std [m/s]	0.08	0.14	0.14	0.09	0.12	0.08
Max(abs) [m/s]	0.41	10.09	10.41	0.91	5.37	0.29

**Table 5 sensors-23-02141-t005:** Attitude angle difference statistics from KVH unit—Case Study 1.

	*ϕ_PBCS_*	*ϕ_LLI_*	*ϕ_MW_*	*ϕ_TE_*
Mean [deg]	0.00	0.00	0.30	0.03
Std. [deg]	0.57	0.57	0.54	0.61
Max(abs) [deg]	1.70	1.70	1.73	1.69
	*θ_PBCS_*	*θ_LLI_*	*θ_MW_*	*θ_TE_*
Mean [deg]	−0.27	−0.27	0.12	−0.23
Std. [deg]	0.38	0.38	0.51	0.42
Max(abs) [deg]	1.59	1.59	1.58	1.58
	*ψ_PBCS_*	*ψ_LLI_*	*ψ_MW_*	*ψ_TE_*
Mean [deg]	−0.12	−0.12	−0.07	−0.12
Std. [deg]	0.22	0.22	0.23	0.22
Max(abs) [deg]	0.83	0.83	0.80	0.80

**Table 6 sensors-23-02141-t006:** Ambiguity resolution success ratios of the different solutions—Case Study 2.

	PBCS	LLI	MW	TE	UBX	KVX
*Bl*_1_ AR [%]	75.34	79.08	33.08	23.06	92.96	0.0
*Bl*_2_ AR [%]	95.44	95.87	4.99	4.99	-	-
*Bl*_3_ AR [%]	93.91	91.70	14.92	7.72	-	-

**Table 7 sensors-23-02141-t007:** Coordinate differences from u-blox moving base receiver—Case Study 2.

	*N_PBCS_*	*N_LLI_*	*N_MW_*	*N_TE_*	*N_KVH_*
Fix. Mean [m]	0.00	0.00	0.00	−0.01	0.00
Fix. Std. [m]	0.01	0.01	0.15	0.32	0.00
Fix. Max(abs) [m]	0.25	0.23	1.67	7.70	0.00
Flo. Mean [m]	0.12	0.14	0.49	0.97	0.98
Flo. Std. [m]	0.15	0.15	3.14	3.23	0.69
Flo. Max(abs) [m]	0.33	0.33	11.44	11.95	2.89
	*E_PBCS_*	*E_LLI_*	*E_MW_*	*E_TE_*	*E_KVH_*
Fix. Mean [m]	0.00	0.00	−0.02	−0.01	0.00
Fix. Std. [m]	0.02	0.02	0.57	0.38	0.00
Fix. Max(abs) [m]	0.22	0.22	15.62	7.23	0.00
Flo. Mean [m]	0.04	0.05	−3.21	−2.97	−1.51
Flo. Std. [m]	0.17	0.18	4.66	4.78	2.06
Flo. Max(abs) [m]	0.42	0.42	20.01	23.65	8.36
	*D_PBCS_*	*D_LLI_*	*D_MW_*	*D_TE_*	*D_KVH_*
Fix. Mean [m]	0.00	0.00	0.11	0.09	0.00
Fix. Std. [m]	0.03	0.03	0.57	0.62	0.00
Fix. Max(abs) [m]	0.18	0.18	11.06	9.5	0.00
Flo. Mean [m]	0.10	0.12	4.06	3.99	−0.56
Flo. Std. [m]	0.13	0.12	3.76	4.28	2.34
Flo. Max(abs) [m]	0.42	0.42	15.04	15.2	5.88

**Table 8 sensors-23-02141-t008:** Horizontal and vertical velocity statistics—Case Study 2.

	*H_PBCS_*	*H_LLI_*	*H_MW_*	*H_TE_*	*H_UBX_*	*H_KVH_*
Mean [m/s]	4.23	4.23	4.64	4.64	4.24	4.25
Std [m/s]	2.84	2.84	3.66	3.35	2.84	2.84
Max(abs) [m/s]	13.27	13.27	71.71	51.81	13.20	13.22
	*V_PBCS_*	*V_LLI_*	*V_MW_*	*V_TE_*	*V_UBX_*	*V_KVH_*
Mean [m/s]	0.04	0.04	0.04	0.04	0.04	0.03
Std [m/s]	0.21	0.21	1.73	1.66	0.23	0.23
Max(abs) [m/s]	1.76	1.76	39.42	36.27	2.81	1.10

**Table 9 sensors-23-02141-t009:** Attitude angle difference statistics from KVH unit—Case Study 2.

	*ϕ_PBCS_*	*ϕ_LLI_*	*ϕ_MW_*	*ϕ_TE_*
Mean [deg]	−0.07	−0.06	−33.25	−33.23
Std. [deg]	0.70	0.70	46.53	55.01
Max(abs) [deg]	2.11	2.11	170.03	179.96
	*θ_PBCS_*	*θ_LLI_*	*θ_MW_*	*θ_TE_*
Mean [deg]	−0.19	−0.18	27.43	26.83
Std. [deg]	0.64	0.64	27.16	28.59
Max(abs) [deg]	3.12	3.12	82.21	83.45
	*ψ_PBCS_*	*ψ_LLI_*	*ψ_MW_*	*ψ_TE_*
Mean [deg]	0.01	0.03	7.49	16.04
Std. [deg]	0.58	0.58	43.70	51.93
Max(abs) [deg]	1.85	1.85	134.36	139.34

**Table 10 sensors-23-02141-t010:** Ambiguity resolution success ratios of the different solutions—Case Study 3.

	PBCS	LLI	MW	TE	UBX	KVX
*Bl*_1_ AR [%]	75.44	56.31	44.92	43.4	79.76	48.24
*Bl*_2_ AR [%]	77.96	67.22	20.47	42.14	-	-
*Bl*_3_ AR [%]	79.56	69.91	10.62	45.61	-	-

**Table 11 sensors-23-02141-t011:** Coordinate differences from u-blox moving base receiver—Case Study 3.

	*N_PBCS_*	*N_LLI_*	*N_MW_*	*N_TE_*	*N_KVH_*
Fix. Mean [m]	0.00	0.03	0.01	0.01	0.14
Fix. Std. [m]	0.02	0.25	0.02	0.14	1.07
Fix. Max(abs) [m]	0.15	1.88	0.19	0.70	5.63
Flo. Mean [m]	−0.40	0.73	0.09	0.08	1.53
Flo. Std. [m]	0.98	1.03	0.70	0.70	2.55
Flo. Max(abs) [m]	3.55	3.65	4.82	4.73	5.53
	*E_PBCS_*	*E_LLI_*	*E_MW_*	*E_TE_*	*E_KVH_*
Fix. Mean [m]	−0.01	−0.05	0.00	−0.01	0.09
Fix. Std. [m]	0.03	0.34	0.02	0.07	0.47
Fix. Max(abs) [m]	0.32	2.47	0.16	0.48	2.53
Flo. Mean [m]	0.49	−1.04	−0.47	−0.46	1.20
Flo. Std. [m]	1.62	1.38	0.85	0.84	1.74
Flo. Max(abs) [m]	12.05	9.20	8.30	8.20	12.93
	*D_PBCS_*	*D_LLI_*	*D_MW_*	*D_TE_*	*D_KVH_*
Fix. Mean [m]	−0.01	−0.01	0.04	−0.02	0.29
Fix. Std. [m]	0.09	0.07	0.09	0.13	3.21
Fix. Max(abs) [m]	0.92	0.69	0.93	0.73	16.54
Flo. Mean [m]	−0.36	−0.45	0.16	−0.28	4.87
Flo. Std. [m]	1.05	0.88	0.90	0.87	7.68
Flo. Max(abs) [m]	24.50	24.38	24.87	24.38	34.00

**Table 12 sensors-23-02141-t012:** Horizontal and vertical velocity statistics—Case Study 3.

	*H_PBCS_*	*H_LLI_*	*H_MW_*	*H_TE_*	*H_UBX_*	*H_KVH_*
Mean [m/s]	4.22	4.22	4.25	4.25	4.29	4.22
Std [m/s]	2.98	2.99	3.02	3.02	3.54	3.15
Max(abs) [m/s]	12.41	15.67	27.97	24.13	82.71	52.05
	*V_PBCS_*	*V_LLI_*	*V_MW_*	*V_TE_*	*V_UBX_*	*V_KVH_*
Mean [m/s]	0.03	0.03	0.03	0.03	0.03	0.03
Std [m/s]	0.24	0.25	0.30	0.28	4.76	2.79
Max(abs) [m/s]	2.38	2.40	3.25	4.47	211.11	119.52

**Table 13 sensors-23-02141-t013:** Attitude angle difference statistics from KVH unit—Case Study 3.

	*ϕ_PBCS_*	*ϕ_LLI_*	*ϕ_MW_*	*ϕ_TE_*
Mean [deg]	−0.31	−0.08	0.57	−0.08
Std. [deg]	0.41	0.57	1.58	0.65
Max(abs) [deg]	2.5	1.75	4.86	2.44
	*θ_PBCS_*	*θ_LLI_*	*θ_MW_*	*θ_TE_*
Mean [deg]	0.11	−0.05	−0.74	−0.31
Std. [deg]	0.32	0.43	1.02	0.47
Max(abs) [deg]	1.48	1.38	5.12	2.03
	*ψ_PBCS_*	*ψ_LLI_*	*ψ_MW_*	*ψ_TE_*
Mean [deg]	0.26	0.27	3.33	0.31
Std. [deg]	0.51	0.54	20.18	0.65
Max(abs) [deg]	2.15	2.49	59.25	2.55

## Data Availability

All of the raw measurement data are available at https://nextcloud.sztaki.hu/s/fQTZwNHkgCm92j7 (accessed on 29 January 2023).

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
