# Peer review of "Position and Attitude Determination in Urban Canyon with Tightly Coupled Sensor Fusion and a Prediction-Based GNSS Cycle Slip Detection Using Low-Cost Instruments"

_sensors, 2023, doi:10.3390/s23042141_

Round 1

Reviewer 1 Report

This paper constructs a moving platform position and attitude estimation algorithm based on the tightly coupled integration of low-cost GNSS, inertial, magnetic and barometric sensors, and proposes a new cycle slip method for GNSS based on prediction. In this method, the triple-differenced carrier phase observation predicted by EKF is used to compare with the actual observed triple-differenced carrier phase observation, so as to determine whether cycle slip occurs. The performance of the proposed algorithm is then evaluated with the vehicle experimental data collected in urban canyon. The paper has a certain reference value. But there are still some aspects that need to be improved. It is suggested to review again after resubmission. The advice of this paper is as follows:

1. In equation (13), the author does not explain the definition of parameter m, please add the corresponding content.

2. How is the threshold value in equation (18) determined? Please explain it.

3. Generally, multi-frequency GNSS receivers cannot be called low-cost equipment. Single frequency GNSS receivers are usually considered as low-cost receiving devices, and obviously, the above conditions are not satisfied in this paper.

4. The LLI-based method used in this paper is usually not the preferred method for cycle slip detection, and the comparison between the proposed algorithm and this method can not fully reflect the advantages of the proposed algorithm. Therefore, it is suggested that the authors should add the comparison experiment results between the proposed algorithm and classical methods such as high-order difference method, MW (Melbourne-Wubeena combination) method or the TurboEdit method et al.

5. In this paper, the proposed prediction-based cycle-slip detection algorithm mainly relies on the comparison between the triple-differenced carrier phase observation based on EKF prediction and the actual triple-differenced carrier phase observation to determine whether cycle slip occurs. However, EKF relies on the dynamic model to predict the system state, and its prediction results are inevitably affected by the inaccuracy of the dynamic model. Therefore, how does the author consider the impact of inaccurate EKF prediction on the proposed cycle slip detection algorithm in this paper?

6. In this paper, the content of multi-sensor tightly-coupled integration algorithm is too simple, and it is recommended to supplement this part.

Author Response

Dear Reviewers and Associate Editor,
Thank you for your insightful comments on our submission. Your feedback has proven valuable in our efforts to improve the quality of the paper.  The detailed list of corrections made and the corresponding explanations are included in the attached pdf.

Best Regards,

Bálint Vanek

Reviewer 2 Report

The author should work on his results and add more case studies. A single case study of approx 5.8 minutes is not enough to validate the method.

A comprehensive discussion on the experiment conducted and methodology is required in the paper.

Author Response

(The authors gave the same response as above.)

Round 2

Reviewer 2 Report

Although, the paper has been significantly improved with more experiments however, it is recommended for the author to proof-read it again to improve the english before it gets published.